# Comparison of Learning Transfer Using Simulation Problem-Based Learning and Demonstration: An Application of Papanicolaou Smear Nursing Education

**DOI:** 10.3390/ijerph18041765

**Published:** 2021-02-11

**Authors:** Jeongim Lee, Hae Kyoung Son

**Affiliations:** Department of Nursing, Eulji University, Seongnam City 13135, Korea; jjungim4u@naver.com

**Keywords:** nursing, education, learning transfer, simulation, problem-based learning

## Abstract

The purpose of this study was to compare the effects of simulation problem-based learning (S-PBL), a type of learning that reflects various clinical situations, and demonstration-based learning, a conventional type of learning that shows clinical skill performance, of Papanicolaou smear education on the self-confidence, learner satisfaction, and critical thinking of nursing students. A quasi-experimental control group pretest-posttest design was used. Nursing students who were classified as advanced beginners were randomly allocated to the control group (*n* = 53) or the experimental group (*n* = 52). Students in the control group participated in a conventional demonstration of a Papanicolaou smear, while students in the experimental group participated in S-PBL. The students’ self-confidence, learner satisfaction, and critical thinking were measured via a self-reported questionnaire. Compared with the control group, self-confidence, learner satisfaction, and critical thinking increase significantly more (*p* < 0.001) in the experimental group. S-PBL was found to be an effective strategy for improving learning transfer, applying learned nursing knowledge to simulated nursing situations. Thus, S-PBL is recommended to improve training in nursing education.

## 1. Introduction

Cervical cancer is one of the most prevalent malignancies of the female reproductive tract, and infection with human papillomavirus (HPV) is considered to be the primary cause [1]. Globally, it is the second most commonly diagnosed carcinoma in women; 569,847 new cases and 311,400 deaths were reported in 2018 [2]. In South Korea, the number of patients diagnosed with cervical cancer has gradually increased from 59,910 in 2017 to 62,071 in 2018 and 63,051 in 2019 [3]. The incidence of cervical cancer has not been decreased dramatically considering the development of improved research tools and test methods. Accordingly, the National Cancer Screening Project in South Korea has promoted the Papanicolaou smear (Pap smear) as a screening method for cervical dysplasia and carcinoma among women over the age of 20 [4].

A Pap smear, introduced by George Papanikolaou in 1943, is a simple, quick, and inexpensive screening procedure for cervical cancer [5]. A Pap smear can be performed in an obstetrics and gynecology outpatient clinic at a medical institution that conducts national cancer screening [1], and in most cases, hospitalization is not required. Therefore, nursing students have limited opportunities to observe Pap smears in maternal nursing practice, which are mainly held in the delivery room and obstetrics and gynecology wards.

In recent years, the demand for professional knowledge and clinical skill performance of nursing students has increased, while participatory observation and direct nursing experiences have become more restricted due to the reinforcement of patient privacy and safety measures. Therefore, simulation-based learning is increasingly being used as an alternative [6,7].

Simulation-based learning can effectively reproduce clinical nursing settings and improve nursing performance through lifelike nursing experiences [8,9]. In particular, simulation problem-based learning (S-PBL) is an educational method that enables nursing students to identify problems in given scenarios and repeatedly perform nursing procedures with a human patient simulator or human body model to improve their nursing competency [7,10].

Previous studies have evaluated simulation of maternal nursing situations, such as delivery or postpartum hemorrhage, and compared the effects of simulation-based learning and clinical nursing practice on clinical performance and learner satisfaction [11,12]. While previous studies have assessed the effects of simulation-based learning in obstetric care, there is a lack of studies on simulation-based learning for preventative screening in gynecological nursing, particularly in performing Pap smears.

Therefore, in this study, we developed Pap smear nursing scenarios based on Jeffries’s simulation model and evaluated the effects on nursing education (Figure 1) [13]. The model is based on constructivist learning theory, and it can be applied in various ways to nursing education as it presents components that can systematically explain nursing simulation education design [14]. Jeffries’s simulation model presented a nursing simulation education design and explained that simulation experiences occur considering design and background in an overall context [13]. As dynamic teacher–student interactions are crucial, teachers need to apply proper simulations based on teacher–student clinical practice to induce educational outcomes. The main components of a simulation model are students, teachers, educational practice, intervention, and expected outcomes [13]. In this study, nursing students are students, faculty is teacher, maternal nursing educational practice is educational practice, S-PBL or demonstration is intervention, and self-confidence, learner satisfaction, and critical thinking are expected outcomes.

A well-developed simulation intervention results in self-confidence, learner satisfaction, and clinical thinking according to Jeffries’s simulation model [13]. A demonstration is a popular way to present skills for nursing students in South Korea. Simulation practice allows students to build their own knowledge system while experiencing virtual situations. As Bandura’s self-efficacy theory [15] self-efficacy leads from knowledge to practice. Self-efficacy is required for students to apply the information they have acquired to the actual situation, and what kind of experience students have at this time is a major factor in determining their self-efficacy. Enactive attainment in simulation education is a major element leading to self-efficacy, and while vicarious experiences (e.g., demonstrations) also lead to self-efficacy, enactive attainment in simulation education is a more important element than vicarious experience.

Self-confidence and learner satisfaction are important aspects of simulation experiences. Such experiences can be offered to provide clinical nursing information for Pap smears. Nurses must identify complex situations and detect each patient’s needs in the midst of substantial information during clinical practice. Thus, nursing students need to develop critical thinking skills to accurately make decisions in complex situations. In Jeffries’s model, the outcome consists of self-confidence, learner satisfaction, critical thinking, and so on [13]. Those results correspond to the skills that the students need to gain from the education.

### 1.1. Aims

S-PBL for Pap smear education provides nursing students with opportunities for interaction and communication with patients, guardians, and medical staff through peer role-play. It promotes teamwork and allows students to think and cope with nursing scenarios by directly applying women’s health nursing knowledge [10]. Therefore, this study implemented S-PBL to assess its effects on self-confidence, learner satisfaction, and critical thinking and compared them to the effects of conventional demonstrations to provide a basis for active learning transfer among nursing students.

More specifically, the objectives of this study were as follows:

First, we assessed self-confidence, learner satisfaction, and critical thinking among nursing students. Next, we evaluated the relationship between self-confidence, learner satisfaction, and critical thinking. Finally, we compared the effects of S-PBL to those of conventional demonstrations.

### 1.2. Research Hypotheses

The hypotheses of this study are as follows.

**Hypothesis** **1:***The S-PBL group will show a higher increase in self-confidence than the control group after the intervention*.

**Hypothesis** **2:***The S-PBL group will show a higher increase in learner satisfaction than the control group after the intervention*.

**Hypothesis** **3:***The S-PBL group will show a higher increase in critical thinking than the control group*.

## 2. Materials and Methods

### 2.1. Research Design

This study utilized a quasi-experimental control group pre-test–post-test design, which is shown in Figure 2. The experimental group engaged in S-PBL based on Pap smear knowledge and the control group participated in a Pap smear demonstration based on Pap smear knowledge, and the two groups’ self-confidence, learner satisfaction, and critical thinking skills were then compared (Figure 2).

### 2.2. Subjects and Setting

Third-year nursing students from a nursing college at a university located in S city (a metropolitan area) in South Korea who participated in a maternal nursing practice course were included in the study. All participants had completed a pre-requisite maternity nursing course and had basic knowledge related to women’s health nursing before the study.

The minimum sample size was calculated using G*Power Software (G * Power 3.1.7, Heinrich-Heine-University, Düsseldorf, Germany) with a significance level of α = 0.05, a power of 0.95, and an effect size of 0.8 [16,17], and 84 subjects with 42 subjects each in the control and experimental groups were required. A total of 105 subjects were included to compensate for the dropout rate of about 20% in data collection. The subjects were assigned to the control group (*n* = 53) and experimental group (*n* = 52) through randomized blinded allocation. No subjects dropped out, and a total of 105 subjects were included in the final analysis, satisfying the minimal required sample size.

### 2.3. Research Procedures

#### 2.3.1. Preliminary Assessment

General characteristics and knowledge about Pap smears were assessed before the intervention for both the experimental and control groups. Knowledge about Pap smears was assessed using 10 items regarding intended targets, purpose, timing, procedures, cautions, and an overall understanding of the anatomical physiology of the female genitalia (i.e., the cervical cancer screening conducted by the National Health Insurance Service is recommended once every two years for women over the age of 20; one can have a Pap smear during menstruation; cervical dysplasia can be detected from the results of Pap smears, etc.). The items were evaluated using a “True or False” quiz format. This tool was developed by two professors with more than five years of experience in maternal nursing instruction, practical education, and clinical experience in maternal nursing. The total score ranged from 0 to 10, and a higher score indicated a higher level of knowledge.

#### 2.3.2. Interventions: Simulation Problem-Based Learning (S-PBL) vs. Demonstration

The experimental group participated in S-PBL in the simulation room of the nursing college. Simulation scenarios were developed by the researchers based on the Korean Society of Obstetrics and Gynecology, literature on women’s health nursing, and clinical cases of Pap smear nursing observed in obstetrics and gynecology departments. A professor with extensive experience in simulation-based maternal nursing was consulted. S-PBL was formatted as a small group activity, consisting of 3–4 students (each student played the role of one doctor, one or two nurse(s), or one guardian) who were randomly selected. According to Benner [18], Pap smear S-PBL is a practical form of education for advanced beginners who are at least second-year students. The difficulty level of the Pap smear S-PBL is Level II intermediate, which means that the students have to consider the individual characteristics of patients and provide nursing based on communication rather than show a mere replay of skills in a specific situation [7]. Pap smear S-PBL consisted of 60 min of pre-briefing, 15–20 min of running time, and 60 min of debriefing, using a human body model with the anatomy of the female genitalia including the cervix (Figure 3). The instructors provided sufficient time to review the knowledge and skills required for orientation and simulation in the pre-briefing session [19]. In the running time, the students performed Pap smear nursing duties, while the instructors who facilitated the simulation provided cues about the Pap smear nursing situation and patient information. In addition, the instructors measured subjects’ Pap smear skill performance using a structured checklist that was developed by two professors who had more than five years of experience in maternal nursing teaching, practical education, and clinical experience in maternal nursing (Table 1). The checklist consisted of 17 items, which were evaluated on a 3-point scale (1 = “not performed,” 2 = “partially performed,” and 3 = “well performed”). Pap smear skill performance score ranged from 17 points to 51 points. In the debriefing stage, subjects provided feedback and reflected on the small group activities.

The control group watched Pap smear demonstrations by the instructors for 15–20 min in groups of 3–4 students and asked questions at the end.

#### 2.3.3. Outcome Assessment

After the intervention, self-confidence, learner satisfaction, and critical thinking were evaluated, using a structured questionnaire to measure learning transfer related to Pap smears, both for the experimental and control group. The 11-point Numeric Rating Scale (NRS), for which the question is “How much self-confidence do you have in Pap smear nursing?” was used to measure the participants’ degree of self-confidence on a scale of 0–10 from “not at all” (0 points) to “very much” (10 points), and a higher score indicates higher self-confidence. The NRS, which is likely the most commonly used self-report tool in various disciplinary fields such as clinical, research, and so on, allows for quick measurement, and NRS has strong psychometric properties, thereby offering a valid approach to the self-assessment of confidence in examination skills by health care students [20,21]. Learning satisfaction was measured with 24 items developed by Yoo [22]. The items (i.e., “I became more interested in this field after taking this class,” “The teaching method made it easy to understand,” etc.) were evaluated on a 5-point Likert scale from “not at all” (1 point) to “very much” (5 points), and a higher score indicated higher learner satisfaction. The reliability of the tool, indicated by Cronbach’s α, was 0.94 in the original paper and 0.94 in our study. Permission was granted by the original author to use the tool in this study [22]. Critical thinking referred to personal dispositions and habits used for problem solving and decision-making. In this study, critical thinking was evaluated using a tool developed by Yoon for nursing [23]. The tool consisted of 27 items (i.e., “When dealing with a complex problem, I judge and deal with the problem according to the criteria I set,” “I withhold my judgment and ponder the matter until adequate and sufficient evidence is obtained,” etc.) that were evaluated on a 5-point Likert scale from “not at all” (1 point) to “very much” (5 points). Items 4 and 14 were reverse-scored, and the total score of the tool ranged from 27 to 135. A higher score indicated better critical thinking. The reliability of the tool, which is indicated by Cronbach’s α, was 0.84 in the original paper and 0.90 in our study [23]. Permission was granted by the original author to use the tool in this study.

### 2.4. Data Collection

This study was conducted from 19 October 2020, to 4 December 2020. The subjects were blinded to the group to which they belonged, and the control group and experimental group participated in the study at different times to prevent spillover effects between the two groups. The researcher provided a sufficient explanation of the research, intervention, and procedure for the research, distributed the questionnaire before and after each intervention, and collected the questionnaire separately from the consent form. Data were acquired using a structured questionnaire, which required approximately 10 min to complete for both the pretest and posttest, and the participants were provided with small compensation (school supplies worth 1000 KRW) for their participation.

### 2.5. Statistical Analysis

Statistical analysis was performed using IBM SPSS Statistics ver. 22.0 program (IBM Co., Armonk, NY, USA). Descriptive statistics were used to analyze the general characteristics of the subjects. Cronbach’s α, Pearson correlation coefficient, and *t*-test were used to analyze the reliability of the evaluation tools, correlation analysis between the variables, and homogeneity test and comparison of intervention effects, respectively. The correlation between the factors was assessed using the correlation coefficient and a *p*-value < 0.05 was considered statistically significant.

### 2.6. Ethical Considerations

This study was approved by the Institutional Review Board of Eulji University (No. EUN20-041). The details of the study were sufficiently explained to the subjects, and written consent was acquired from those who voluntarily wished to participate. The subjects were allowed to withdraw from the study at any time and for any reason, and it was explained that there would be no disadvantages if they refused to participate, to relieve any unnecessary tensions in the study. The control group was provided with an opportunity to take part in S-PBL without data collection after the study was completed.

## 3. Results

### 3.1. Homogeneity Test

Table 2 shows the general characteristics and homogeneity test of variables between the experimental and control groups. The mean age (standard deviation [SD]) of subjects in the experimental and control groups was 22.54 (2.84) and 22.08 (1.99), respectively. The experimental group consisted of 41 female students (78.8%) and 11 male students (21.2%), while the control group consisted of 44 female students (83.0%) and 9 male students (17.0%). Subjects generally make an independent choice to study nursing, with 32 (61.5%) and 23 (43.4%) in the experimental and control groups, respectively, and 37 (71.2%) and 40 (75.5%) students were very satisfied or satisfied with their major in the experimental and control groups, respectively. The number of students who were satisfied with clinical nursing practice was the greatest in both groups, with 23 (44.2%) and 28 (52.8%) in the experimental and control groups, respectively. The number of students who were moderately satisfied with academic achievement was the greatest, with 32 (61.5%) and 21 (39.6%) in the experimental group and control group, respectively, and 36 (69.3%) students in the experimental group and 31 (58.5%) students in the control group had high or very high academic stress. Preliminary assessment of Pap smear knowledge showed that there was no significant difference between the two groups, with a mean score (SD) of 6.75 (1.55) points and 6.62 (1.44) points in the experimental group and control group, respectively (t = 0.43, p = 0.127). Pre-test homogeneity tests between the two groups showed that the general characteristics, self-confidence (t = 0.51, *p* = 0.612), learner satisfaction (t = 0.72, *p* = 0.475), and critical thinking (t = 1.42, *p* = 0.158) were homogeneous (*p* > 0.05).

### 3.2. Correlation between Variables

Correlations between the variables are shown in Table 3. Self-confidence was positively correlated with learner satisfaction (r = 0.361, *p* < 0.001) and critical thinking (r = 0.208, *p* = 0.033), and learner satisfaction was positively correlated with critical thinking (r = 0.622, *p* < 0.001).

### 3.3. Effects of Intervention

The effects of the Pap smear S-PBL and demonstrations were compared. Self-confidence, learner satisfaction, and critical thinking were significantly improved in both groups post-intervention (experimental group: self-confidence t = 10.52, *p* < 0.001, learner satisfaction t = 9.50, *p* < 0.001, and critical thinking t = 5.09, *p* < 0.001; control group: self-confidence t = 10.84, *p* < 0.001, learner satisfaction t = 6.58, *p* < 0.001, and critical thinking t = 4.36, *p* < 0.001) (Table 4).

When the post-intervention effects were compared between the two groups, it was observed that self-confidence (t = 2.52, *p* = 0.013), learner satisfaction (t = 3.47, *p* = 0.001), and critical thinking (t = 2.07, *p* = 0.041) had improved significantly more in the experimental group than in the control group (Table 5). Therefore, hypothesis 1, “The S-PBL group will show a higher increase in self-confidence than the control group after the intervention,” hypothesis 2, “The S-PBL group will show a higher increase in learner satisfaction than the control group after the intervention,” and hypothesis 3, “The S-PBL group will show a higher increase in critical thinking than the control group,” are all supported.

## 4. Discussion

In this study, Pap smear nursing S-PBL was developed based on Jeffries’ simulation model, and its effects on self-confidence, learner satisfaction, and critical thinking were assessed and compared to the effects of a conventional demonstration [13].

Self-confidence was found to have significantly increased in the experimental group than in the control group after the intervention. Accurate self-evaluation of a learner’s nursing skill performance is a necessary step for professional nurses to understand their strengths and weaknesses and to promote their development [24]. Simulations provide nursing students with opportunities to practice decision-making and develop team skills in a non-threatening environment [25]. These findings are consistent with the results of previous studies [26]. Moreover, results of simulations of rapidly deteriorating clinical cases with medical students [27] and preoperative nursing skills simulations with nursing students are also consistent with the findings of our study [28].

In contrast, in a study by Brannan, White, and Bezanson [29], there was no significant difference in self-confidence after traditional classroom lectures and instructional methods using the human patient simulator as a tool for experiential learning. It is possible that differences in learning design, such as pre-briefing and debriefing time, and number of students per group affected the results. In particular, the current study was designed such that communication between nurse, doctor, and patient could be performed through peer role-play, to encourage appropriate nursing behaviors. In simulations, communication and emotion affect confidence [30]. Therefore, it is necessary to consider the effects of positive interactions and emotions on self-confidence in the experimental group who took part in S-PBL. In future studies, qualitative evaluation of communication and emotion in students, quantitative evaluation of the relationship between these factors, and confidence in the context of various educational methods, such as S-PBL and demonstrations, should be explored.

In our study, learner satisfaction increased significantly more in the experimental group than in the control group, and this finding was consistent with that of a previous study [31]. The S-PBL used in this study required the subjects to engage in appropriate communication beyond relaying disease-specific information, therefore their confidence, and thereby their satisfaction with the program, increased [32].

Pap smears are mainly performed in medical checkup centers, and obstetrics and gynecology outpatient departments, so it is difficult for nursing students, who are mainly placed in wards, operating rooms, and delivery rooms, to experience their administration in nursing practice. The program stimulated students’ curiosity and motivated them to participate actively in the class by exposing them to scenarios and skills that they were unable to experience through clinical practice. Preliminary activities, such as orientation and question-and-answer sessions related to simulation, were included to induce active participation. Previous studies have shown that this learner-centered education strategy is useful for nursing students who need to acquire complex nursing knowledge and skills [29]. Therefore, it would be beneficial to stimulate the interest of students through S-PBL about frequently occurring clinical nursing cases that are difficult for nursing students to experience in order to maximize the learning transfer effects for knowledge, skills, and attitudes.

In our study, critical thinking increased significantly more in the experimental group than in the control group. This finding is consistent with the results of various other simulations that involve apnea in high-risk infants, rapport-building, febrile infant care, and delivery nursing [33,34]. The S-PBL was configured such that communication was facilitated by peer role-play and nursing skill practice was facilitated by the use of an anatomical structure model of the female genitalia. Our findings are consistent with previous studies that reported that critical thinking of nursing students was improved through peer role-play-based simulation education, which is widely used in nursing education [34,35]. However, multi-mode simulation with ‘high-fidelity simulator and peer role-play’ or ‘standardized patient and peer role-play’ have been shown to improve critical thinking, communication skills, self-efficacy, and level of understanding patient and guardian perspectives [34,35]. However, it is necessary to verify the effects of peer role-play-based simulation education with various research designs, because this type of education can be used in universities that are unable to use high-fidelity simulators due to high costs.

In this study, sufficient time was provided to the students during the pre-briefing session, which likely increased critical thinking. Pre-briefing time for S-PBL preparation plays a key role in inducing critical thinking and facilitating learning [36]. Thus, it is important to provide sufficient pre-briefing time during simulation practice to encourage students to solve problems on their own.

Significant differences were observed when we compared pre- and post-intervention self-confidence, learner satisfaction, and critical thinking in the two groups of this study. In our study, a question-and-answer session was held with both the control and experimental groups. Interaction of students with instructors and feedback improved learning transfer [37]. Therefore, it is necessary to promote learning transfer through appropriate question-and-answer sessions and feedback in conventional demonstrations in addition to S-PBL. Moreover, demonstrations of medical equipment are effective [19], so conventional demonstrations during pre-briefing sessions may further enhance understanding, decrease unnecessary tension among students, and maximize the learning transfer effects [10,19].

Finally, the results of this study suggest that S-PBL with peer role-play can be an effective alternative for nursing education environments where high-fidelity simulators or trained standardized patients are not accessible due to high costs [10]. In addition, the use of S-PBL is especially useful due to the coronavirus disease 2019 (COVID-19) situation because S-PBL enhances learning transfer, which has resulted in limited clinical practice for nursing students. Simulation learning allows students to take on the role that they will be expected to perform as practicing nurses [37]; therefore simulation level should be set according to learning objectives. Sufficient pre-briefing, demonstrations, and evaluation tools for learning transfer should be used to maximize the effects of S-PBL.

### Limitations

In this study, the effects of short-term interventions for Pap smear education were assessed, and self-confidence, learner satisfaction, and critical thinking were evaluated through self-reported questionnaires to compare intervention effects. However, a cross-sectional study has an inherent limitation, in that causality cannot be determined. The following limitations must be considered in the interpretation and generalization of the results. The learning interventions were based on a single Pap smear scenario using a human body model of the female genitalia. A comparison of the effects of various interventions (e.g., with a high-fidelity simulator or multi-mode simulation) using multiple Pap smear scenarios would provide additional insight about the most effective method of educating nursing students on how to perform Pap smears.

## 5. Conclusions

Systematic and integrated application of S-PBL that can provide learning experiences in various nursing scenarios that are necessary as an alternative to limited direct nursing during clinical practice. Therefore, in this study, a Pap smear S-PBL using a human body model of the female genitalia was developed and applied, and the effects were assessed. We suggest that S-PBL, which can increase self-confidence, learner satisfaction, and critical thinking more than can conventional demonstrations, is an effective alternative educational strategy to compensate for the limitations of clinical nursing practices. Furthermore, we recommend that S-PBL be utilized more frequently when developing nursing curricula. In addition, we provided basic data to maximize the effects of learning transfer. In particular, this study is significant in terms of theoretical implication in that it confirmed the concept of learning transfer and verified its effectiveness based on Jeffries’s simulation model. As this theory is concise, clear, and easy to apply [14], it can be used as a basis for designing simulation education for other subjects. As shown in the results of this study, Jeffries’s model has clinical significance as it can enhance the ability to actively cope with changes in the environment and circumstances, thereby improving the students’ competence for solving patients’ health problems. In terms of implications for practice in particular, the learning transfer effect of S-PBL when applied to various clinical cases will help nursing students provide specialized care in various situations as nurses in the future. Finally, in order to overcome the limitations of clinical practice in situations such as the COVID-19 pandemic, the results suggest that reproducing a realistic environment through a nursing simulation based on various cases can be useful as a teaching method that enables learners to have an authentic experience with active involvement.

## Figures and Tables

**Figure 1 ijerph-18-01765-f001:**
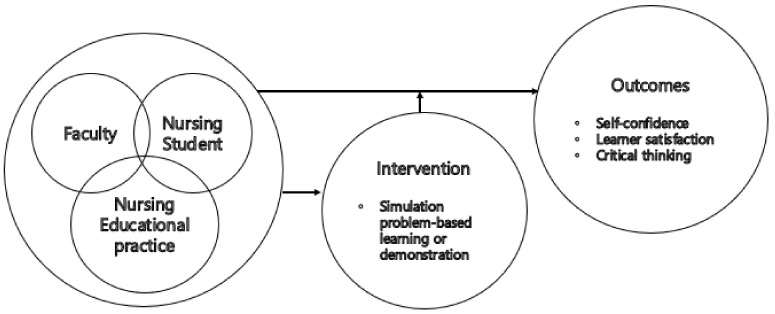
Research framework based on simulation model.

**Figure 2 ijerph-18-01765-f002:**
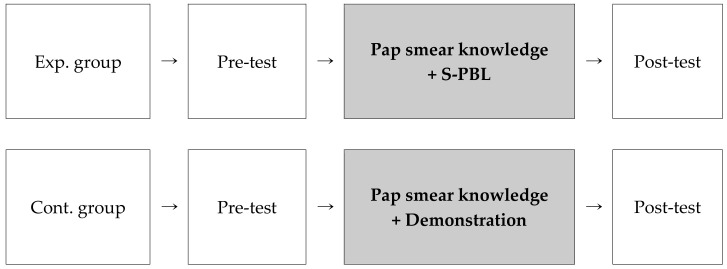
Research design. Exp. group = experimental group, Cont. group = control group; S-PBL = simulation problem-based learning.

**Figure 3 ijerph-18-01765-f003:**
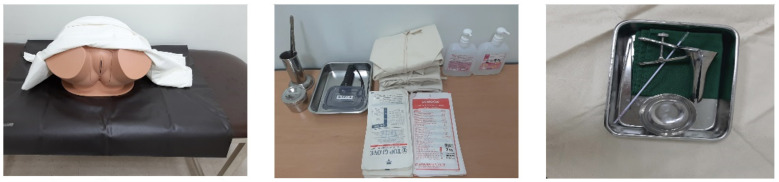
Preparations of Pap smear simulation problem-based learning.

**Table 1 ijerph-18-01765-t001:** Checklist of items of Pap smear simulation problem-based learning.

Category	Item
Nursing scenario	A patient (F/20 years old) visited an obstetrics and gynecology outpatient clinic with her mother for cervical cancer screening. Provide Pap smear nursing to the patient who is worried about her first gynecological examination.
Attitude	1. Wash hands with water and soap
	2. Introduce yourself to the patient
	3. Check who is the subject of examination between the patient and guardian
	4. Ask the patient’s name with an open-ended question to confirm the patient and verify the patient (name, registration number) by comparing hospitalization ID bracelet with the patient list (or prescription form)
	5. Provide an explanation before every treatment
	6. Listen to the patient and answer the questions accurately and clearly
Assessment	7. Assess information related to Pap smear (e.g., regular check-ups, sexual experience, health history, obstetric history, etc.)
	8. Assess whether it is possible to test for cervical cancer today (e.g., vaginal insertion, vaginal washing, sexual interaction, menstruation, etc.)
	9. Assess patient’s anxiety
Skill	Perform treatment based on the assessed information
	10. Provide information on Pap smear examination (e.g., undressing and changing into skirt for examination, lithotomy position, etc.)
	11. Protect the privacy of the patient (e.g., covering the examination area with a screen or sheet, etc.)
	12. Provide emotional support to the patient
	13. Perform aseptic techniques
	14. Perform Pap smear test
	15. Explain to the patient about the current situation and how the results of the test will be delivered
Communication	16. When necessary, communicate effectively with multi-disciplinary team members (nurse or doctor) (e.g., Situation-Background-Assessment-Recommendation)
	17. Update records (e.g., Overall condition of the patient, nursing performance, etc.)

**Table 2 ijerph-18-01765-t002:** Homogeneity test for participants’ general characteristics and variables.

Characteristics	Categories/Range	Mean (SD ^3^)/Frequency (%)		t/*p*
		Exp. ^1^ (*n* = 52)	Cont. ^2^ (*n* = 53)	
age (years)		22.54 (2.84)	22.08 (1.99)	0.97 (0.335)
gender	female	41 (78.8)	44 (83.0)	0.54 (0.590)
	male	11 (21.2)	9 (17.0)	
motivationfor major choice	employment	12 (23.1)	18 (34.0)	1.72 (0.088)
	other’s recommendation	8 (15.4)	12 (22.6)	
	one’s own will	32 (61.5)	23 (43.4)	
satisfaction with major	very satisfied	8 (15.4)	9 (17.0)	0.83 (0.409)
	satisfied	29 (55.8)	31 (58.5)	
	moderate	13 (25.0)	13 (24.5)	
	unsatisfied	1 (1.9)	0 (0.0)	
	very unsatisfied	1 (1.9)	0 (0.0)	
clinical practice satisfaction	very satisfied	4 (7.7)	7 (13.2)	1.93 (0.057)
	satisfied	23 (44.2)	28 (52.8)	
	moderate	18 (34.6)	16 (30.2)	
	unsatisfied	7 (13.5)	2 (3.8)	
	very unsatisfied	0 (0.0)	0 (0.0)	
academic achievement	very high	2 (3.8)	7 (13.2)	0.94 (0.348)
	high	12 (23.1)	14 (26.4)	
	moderate	32 (61.5)	21 (39.6)	
	Low	3 (5.8)	10 (18.9)	
	very low	3 (5.8)	1 (1.9)	
Academic stress	very high	11 (21.2)	6 (11.3)	1.57 (0.119)
	high	25 (48.1)	25 (47.2)	
	moderate	11 (21.2)	14 (26.4)	
	Low	5 (9.6)	7 (13.2)	
	very low	0 (0.0)	1 (1.9)	
learning knowledge		6.75 (1.55)	6.62 (1.44)	0.43 (0.127)
self-confidence		4.48 (2.50)	4.72 (2.26)	0.51 (0.612)
learner satisfaction		102.08 (11.06)	100.51 (11.32)	0.72 (0.475)
critical thinking		105.92 (11.56)	102.81 (10.84)	1.42 (0.158)

^1^ Exp. = experimental group, ^2^ Cont. = control group, ^3^ SD = standard deviation.

**Table 3 ijerph-18-01765-t003:** Correlation between variables.

	r (*p*)		
	Self-Confidence	Learner Satisfaction	Critical Thinking
self-confidence	1	0.361 (<0.001) **	0.208 (0.033) *
learner satisfaction		1	0.622 (<0.001) **
critical thinking			1

* *p* < 0.05, ** *p* < 0.01.

**Table 4 ijerph-18-01765-t004:** Comparison of pretest and posttest self-confidence, learner satisfaction, and critical thinking.

Characteristics	Mean (SD ^3^)					
	Exp. ^1^ (*n* = 52)			Cont. ^2^ (*n* = 53)		
	Pre-Test	Post-Test	t/*p*	Pre-Test	Post-Test	t/*p*
self-confidence	4.48 (2.50)	7.94 (1.59)	10.52 (<0.001) **	4.72 (2.26)	7.15 (1.63)	10.84 (<0.001) **
learner satisfaction	102.08 (11.06)	114.58 (7.18)	9.50 (<0.001) **	100.51 (11.32)	108.68 (10.05)	6.58 (<0.001) **
critical thinking	105.92 (11.56)	111.37 (12.67)	5.09 (<0.001) **	102.81 (10.84)	106.45 (11.67)	4.36 (<0.001) **

** *p* < 0.01, ^1^ Exp. = experimental group, ^2^ Cont. = control group, ^3^ SD = standard deviation.

**Table 5 ijerph-18-01765-t005:** Comparison of self-confidence, learner satisfaction, critical thinking between the experimental and control groups.

Characteristics	Exp. ^1^ (*n* = 52)	Cont. ^2^ (*n* = 53)	t/*p*
	Mean (SD ^3^)	Mean (SD)	
self-confidence	7.94 (1.59)	7.15 (1.63)	2.52 (0.013) *
learner satisfaction	114.58 (7.18)	108.68 (10.05)	3.47 (0.001) **
critical thinking	111.37 (12.67)	106.45 (11.67)	2.07 (0.041) *

* *p* < 0.05, ** *p* < 0.01, ^1^ Exp. = experimental group, ^2^ Cont. = control group, ^3^ SD = standard deviation.

## Data Availability

The data presented in this study are available on request from the corresponding author. The data are not publicly available due to restrictions eg privacy or ethical.

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
