# Peer review of "Comparison of Learning Transfer Using Simulation Problem-Based Learning and Demonstration: An Application of Papanicolaou Smear Nursing Education"

_ijerph, 2021, doi:10.3390/ijerph18041765_

Round 1
Reviewer 1 Report
- The purpose of this study was to compare the effects of simulation problem-based learn-8 ing (S-PBL), a type of learning that reflects various clinical situations, and demonstration-based 9 learning, a conventional type of learning that shows clinical skill performance, of Papanicolaou 10 smear education on the self-confidence, learner satisfaction, and critical thinking of nursing stu
- Please explain Jeffrie’s simulation model in a more detailed way. In my opinion, the description you offer is too summarized.
- Why did you choose self-confidence, learner satisfaction, and critical thinking as outcomes of the implementation of S-PBL?
- You need to offer theoretical background to support your hypotheses.
- Please describe figures 1 and 2 in the text.
- What measure did you use to assess knowledge about Pap smears? Please provide some items sample.
- The same with the remaining measures. Please provide additional information regarding the measures.
- Please characterize both samples.
- The results section should include information regarding your hypotheses. This section should be much more developed.
- Please develop the theoretical implications of the research.
- The implications for practice should also be more developed and clearly stated.
Author Response
Thank you for your considerate opinions. Each review opinion has been fully considered, as shown below, to improve the quality of our research paper. We hope you are staying healthy and safe in the midst of the COVID-19 pandemic. Thank you.

Reviewer 2 Report
I appreciate the opportunity to review this work. The topic is very interesting, comparing the effects of learning based on simulation problems (S-PBL) and learning based on demonstrations. This last methodology is very important in the training of future nurses. I congratulate the authors for this study and I suggest some considerations to try to improve the article.
It would be useful to add more information about the study population that sets the educational context. I do not intend to identify the sample, but if I contextualize the study, the authors speak of the city S, is it possible to know more about that city, for example, in which country is it?
Regarding the sample, the authors indicate that they are third-year students, was the intervention made in a particular subject of the nursing degree curriculum?
Figure three includes three images and a table, it would be appropriate to separate the figure (images) on the one hand and the table on the other and name it appropriately in the text for a better understanding. Was the self-confidence scale evaluated with 11 items or with 11 points? is it a validated scale? How can you tell which items were asked?
Another question, the data collection only required 10 minutes? that collected pre p post, can the authors better specify this procedure?
I reiterate my congratulations to the authors for the study developed
Author Response

(The authors gave the same response as above.)

Round 2
Reviewer 1 Report
I am satisfied with the improvements made by the authors.